# From Chestnut Tree (*Castanea sativa*) to Flour and Foods: A Systematic Review of the Main Criticalities and Control Strategies towards the Relaunch of Chestnut Production Chain

Chiara Aglietti, Alessio Cappelli * and Annalisa Andreani

Department of Agriculture, Food, Environment and Forestry (DAGRI), University of Florence, Piazzale delle Cascine, 18-50144 Florence, Italy
* Correspondence: alessio.cappelli@unifi.it

**Abstract:** *Castanea sativa* plays a key role in different production chains (timber, flour, honey, and tannins). Moreover, considering the great importance of chestnuts and chestnut flour for the food industry and for the subsistence of piedmont communities, a relaunch of this production chain is definitely essential, thus motivating this review. The first aim of this literature overview is to summarize current knowledge regarding the main criticalities in chestnut tree cultivation, chestnut processing, and in chestnut flour production. The second aim is to suggest specific improvement strategies to contrast the main pests and diseases affecting chestnut trees, improve chestnut processing and flour production, and, finally, valorize all by-products generated by this production chain. With respect to chestnut trees, it is essential to develop specific integrated strategies based on early detection and management to contrast known and emerging issues. With regard to chestnut drying and flour production, particular attention needs to be paid to molds and mycotoxins which definitely represent the main criticalities. In addition, further investigations are needed to improve the dying process in both traditional and modern dry kilns, and to develop innovative drying processes. Finally, to face the monumental challenge of environmental sustainability, the valorization of the whole chestnut by-products is crucial. This review clearly highlighted that the recovery of polyphenols from chestnut by-products is the most interesting, sustainable, and profitable strategy. However, the fungal fermentation or the incorporation of little amounts of these by-products into foods seems a very interesting alternative.

**Keywords:** chestnuts pests; chestnut tree diseases; chestnuts molds and mycotoxins; chestnut by-products valorization; chestnut drying; chestnuts post-harvest

## 1. Introduction

Chestnut species are of high importance due to the possibility of being domesticated in several parts of the world [1]. Among these, *Castanea mollissima Blum.* (Chinese chestnut), *C. dentata* (Marsh.) *Borkh.* (American chestnut), *C. crenata Sieb. et Zucc.* (Japanese chestnut), and *C. sativa Mill.* (European or sweet chestnut) are the most widely cultivated species [2]. Due to their capacity of spontaneous hybridizing through cross-pollination and to their high genetic variability, *Castanea* species are reported as capable to adapt to different environmental conditions [3]. For this reason, they have been cultivated since ancient times for producing timber, nuts, honey, and tannins, acquiring, successively, other meanings as carriers of ecosystem services (e.g., protection forests against natural hazards) [4]. *C. sativa* is considered to be the species mainly linked to human activities. Even if its optimum range of growth is reported in Mediterranean and Sub-Mediterranean areas (8–15 °C temperature, 600–800 mm minimal annual precipitation), sweet chestnuts can also be found in Atlantic climates where more than 1700 mm rain throughout the year can occur [1].

*Castanea sativa* acquired a central role in Europe where it was cultivated since the Middle Ages both for food as orchards and for timber as coppices, constituting the pri-

mary source of food and income for mountain populations [5]. In addition to productive roles, *C. sativa* leaves and flowers played a key role in medicine treatments [2]. Although European *C. sativa* still cover more than 2.5 million hectares of forest area, chestnut cultivation has declined since the advent of agricultural and industrial revolutions, with the consequent migration from mountain and rural folks to urban areas [4]. For example, the phenomenon of abandonment in Italy brought to a decrease of 90% of chestnuts cultivated ha during the 20th century, reaching 60,000 ha from the initial 608,000 ha [5]. This trend was further accelerated by World War II that, coupled with fluctuations of extreme weather, pests, and pathogens, led to a period of crisis for chestnut cultivation.

According to an UN-FAO report, China is the biggest producer of chestnuts followed by Bolivia, Turkey, Korea, and Italy [6]. Italy is the second biggest sweet chestnut producer in Europe, with 52,000 tons, and a cultivated area of 21,500 ha in 2014 [7]. Fresh chestnut fruits contain approximately 50% water, with 180 calories per 100 g of edible product [8]. Moreover, chestnuts are mainly composed of starch, from 38% up to 80% where 21.5% is rapidly digestible starch, 20.9% is slowly digestible starch, and 57.6% is resistant starch [8]. The free sugar content can be up to one-third of the total sugars. Moreover, several studies revealed the presence of several mono- and disaccharides as well as of fibers [6,9]. Chestnuts contain a very small amounts of fat (approximately 1%) mainly monounsaturated (MUFA) and polyunsaturated (PUFA) fatty acids [6,8]. Additionally, the protein content is relatively low (below 5%), but of high biological value [8]. However, chestnut fruits also contain significant amounts of g-aminobutyric acid, vitamins E, C, B1, B2, B3, pantothenic acid, pyridoxine, and folate. Moreover, they are an important source of minerals such as Ca, P, K, Mg, S, Fe, Cu, Zn, and Mn. Finally, chestnut flour presents high-quality proteins with essential amino acids (4–7 g/100 g), a high amount of sugars (20–30 g/100 g) and starch (50–60 g/100 g), dietary fibre (4–10 g/100 g), and a low amount of fat (2–4 g/100 g), mostly unsaturated [10]. In addition, chestnut flour represents a good source of phenolic compounds, minerals, and vitamins [10].

Given the essential role played by chestnut trees in different production chains, and considering the great importance of chestnuts and chestnut flour in the food industry, the first aim of this review is to summarize current knowledge regarding the main criticalities in chestnut tree cultivation, chestnut processing, and in chestnut flour production in order to rediscover and relaunch the whole production chain. The second aim is to suggest specific improvement strategies to contrast the main pest and diseases affecting chestnuts and chestnut tree, to improve chestnut processing and flour production, and, finally, to valorize all by-products generated by this production chain to increase sustainability, profitability, and product quality.

## 2. Search Strategy

The literature review explored three databases: ScienceDirect, PubMed, and the Web of Science. The search strings used were:

- Chestnut tree AND chestnut AND (pathologies OR infections OR diseases OR control) NOT ("water chestnut" OR "horse chestnut");
- Chestnut AND (insects OR pests) NOT ("water chestnut" OR "horse chestnut");
- Chestnut flour AND (safety OR storage OR pathogens OR toxins OR drying OR milling OR kneading OR baking).

No language, time, year, or publication status restrictions were imposed. Moreover, all duplicates were removed. The initial results were screened by title and abstract reading, and, successively, by a full-text reading. Figures 1–3 summarize, in the form of flow chart, the obtained results for ScienceDirect, PubMed, and the Web of Science.

## ScienceDirect

**a) Chestnut tree AND chestnut AND (pathologies OR infections OR diseases OR control) NOT ("water chestnut" OR "horse chestnut")**

Articles identified through search string (n = 7002)

→ Articles excluded after the reading of title and abstract (n = 6733)

Full articles assessed for full text reading (n = 269)

→ Articles excluded after the reading of the full text (n = 211)

Full text articles assessed for inclusion (n = 58)

**b) Chestnut AND (insects OR pests) NOT ("water chestnut" OR "horse chestnut")**

Articles identified through search string (n = 4218)

→ Articles excluded after the reading of title and abstract (n = 4180)

Full articles assessed for full text reading (n = 38)

→ Articles excluded after the reading of the full text (n = 22)

Full text articles assessed for inclusion (n = 16)

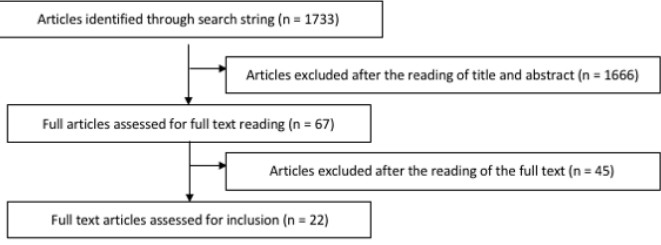

**c) chestnut flour AND (safety OR storage OR pathogens OR toxins OR drying OR milling OR kneading OR baking)**

Articles identified through search string (n = 1733)

→ Articles excluded after the reading of title and abstract (n = 1666)

Full articles assessed for full text reading (n = 67)

→ Articles excluded after the reading of the full text (n = 45)

Full text articles assessed for inclusion (n = 22)

**Figure 1.** Flow charts pertaining to the selection process of papers on ScienceDirect, summarizing the obtained results of the systematic literature review.

## PubMed

**a) Chestnut tree AND chestnut AND (pathologies OR infections OR diseases OR control) NOT ("water chestnut" OR "horse chestnut")**

Articles identified through search string (n = 230)

→ Articles excluded after the reading of title and abstract (n = 119)

Full articles assessed for full text reading (n = 111)

→ Articles excluded after the reading of the full text (n = 101)

Full text articles assessed for inclusion (n = 10)

**b) Chestnut AND (insects OR pests) NOT ("water chestnut" OR "horse chestnut")**

Articles identified through search string (n = 232)

→ Articles excluded after the reading of title and abstract (n = 188)

Full articles assessed for full text reading (n = 44)

→ Articles excluded after the reading of the full text (n = 32)

Full text articles assessed for inclusion (n = 12)

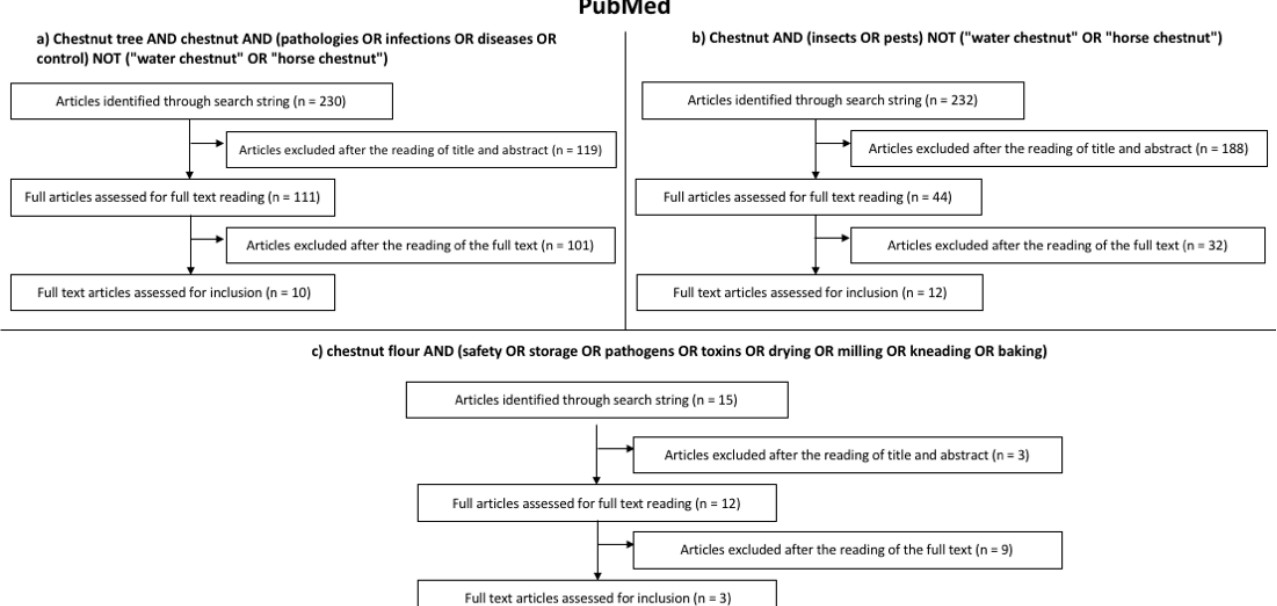

**c) chestnut flour AND (safety OR storage OR pathogens OR toxins OR drying OR milling OR kneading OR baking)**

Articles identified through search string (n = 15)

→ Articles excluded after the reading of title and abstract (n = 3)

Full articles assessed for full text reading (n = 12)

→ Articles excluded after the reading of the full text (n = 9)

Full text articles assessed for inclusion (n = 3)

**Figure 2.** Flow charts pertaining to the selection process of papers on PubMed, summarizing the obtained results of the systematic literature review.

**Web of Science**

**Figure 3.** Flow charts pertaining to the selection process of papers on Web of Science, summarizing the obtained results of the systematic literature review.

## 3. Results of the Systematic Literature Review

The initial dataset consisted of 14,980 items. This was reduced to 196 following the criteria reported in Section 2 (search strategy). The removal of duplicates left a final total of 97 items: 4 book chapters, 6 reviews, and 87 research papers. Figures 1–3 report the results of the selection process for the three databases (ScienceDirect, PubMed, and the Web of Science), consistently with the PRISMA statement [11].

## 4. Chestnut Tree

### 4.1. Pests and Diseases Affecting Chestnut Tree

Climatic stresses, such as severe cold and frost or drought coupled with high temperatures, can affect plant development, facilitating the attack of pests [12]. This is particularly true when dealing with invasive and non-native pests and pathogens, which are representing a growing worldwide problem accompanying environmental changes. Human activities coupled with trade routes are reported as the most responsible of pests and pathogen movements. Nevertheless, climatic, ecological, and environmental conditions are thought to be the major drivers of insects and microbe establishment into new ecosystems. As a consequence, the different conditions found by the introduced organisms in the new environment are able to interact with its key characteristics, limiting or favoring its harmful effects [13,14]. However, both chestnut growth and production have been affected by introduced organisms [12]. This trend has been suggested as a consequence of the absence of natural enemies in the introduction environments that, coupled with the lack of long-term coevolution with hosts founded in the new area, has limited the resistance of native hosts [15].

#### 4.1.1. Ink Disease

Ink disease is caused by oomycetes belonging to the *Phytophthora* genus [16]. The first reports of chestnut root disease causing wilt and tree death were assessed as a consequence of *P. cinnamomi* Rands and *P. cambivora* (Petri) Buisman spread [17]. Little is known about their introduction pathways and origin areas, assuming as possible endemic places, those having mild temperatures and high humidity conditions as some regions of south-east Asia. With respect to *P. cinnamomi*, genetic and population studies have confirmed these hypotheses, identifying regions of Taiwan or New Papua Guinea as the most probable origin

of this species [17]. Ink disease development is in general favored by high temperatures coupled with waterlogging and soil with pH higher than 5.4, being strongly hampered by cold and drought conditions [18].

The consequences of ink disease attack in Spain have caused the disappearance of up to 80% of the original chestnut populations. In addition, these organisms have the ability to survive for years in soil after the infection, making the management and restoration of infected areas particularly challenging [19]. This is reported as a consequence of the *Phytophthora spp.* ability to resist in soil as saprophytes or asymptomatically in plants by exploiting structures as sexual oospores, asexual chlamydospores or intercellular hyphal aggregates. Nowadays, ink disease can be found worldwide, affecting chestnut across its distribution range [18]. The *Phytophthora* species usually associated with ink disease differ based on the geographic area of interest. Consequently, in recent years, new species such as *P. plurivora* T. Jung and T.I Burgess, *P. cryptogea* Pethybridge and Lafferty, *P. citricola* Sawada, *P. cactorum* (Leb. and Cohn) Schröeter and *P. katsurae* W. H. Ko and H. S. Chang, were found to be linked to this disease [20].

### 4.1.2. Chestnut Blight

Fifty years after the ink disease outbreak, a new disease, named chestnut blight, was reported in chestnuts [21]. Its spread was supposed as a consequence of the import of infected chestnut plants from Asia and Japan, many of which were aimed at obtaining plants resistant to ink disease [22]. Symptoms caused by this disease were visible on stems, branches, and twigs, depending on the infection grade, on the *Castanea* species and on the tree age [23]. Typically, when susceptible hosts were infected, necrotic lesions on the bark were developed, creating cankers that were able to expand during years girdling and killing the infected tree part. On the surface of cankers, stromata harboring the fruiting bodies of the disease causal agent were observed [21]. This pathogen is an ascomycete fungus that was first described as *Diaporthe parasitica* Murrill., being later transferred to the *Endorthia* genus and, only in 1978, classified with its current name *Cryphonectria parasitica* (Murr.) Barr. [22].

From its first introduction in the USA, that almost caused the extinction of American chestnut, the pathogen rapidly spread across Europe, with its first detection occurring during 1938 in the Genoa Italian international port [24]. By 1950, it was widespread across chestnut stands of Italy, later colonizing most of southern Europe [25]. Even if the introduction of *C. parasitica* in Europe had impacts on *C. sativa* trees assimilable to those observed on the American chestnut variety, the consequences of its attack in Europe were less severe. In these cases, canker development was slowed down and stopped by the host production of callus tissues that allowed the survival of chestnut cambium, making it possible for the plant to produce new layers of bark under the infected area [26]. The phenomenon was explained starting from the isolation of *C. parasitica hypovirulent* strains from healing cankers in 1965. Subsequent studies assessed, as a cause of their lower virulence, the presence in the fungal cytoplasm of analyzed strains of a double-stranded RNA *hypovirus* belonging to different species that was able to infect these fungi, passing on other *C. parasitica* strains by hyphal anastomosis [27].

The clonal genetic structure observed among the European *C. parasitica* populations, has thought to be one of the major drivers that led to a successful spread of hypovirulence, helped also by the ability of hypoviruses to inhibit fungal sexual reproduction. Therefore, despite the attempt to control chestnut blight using hypoviruses helped the recovery of many European chestnut populations, biocontrol has been unsuccessful in North America, having positive consequences on only isolated populations of *C. dentata* [27]. However, it led the groundwork for the development of American chestnut resistance breeding that has become the major control strategy of chestnut blight. Nowadays, the disease can be found across all the main chestnut-growing areas both in Europe and America [21]. Recently a new *Cryphonectria* species (*Cyphonectria naterciae* Bragança, E. Diogo and A.J.L. Phillips.) was detected on European chestnut trees associated with severe attack of *C. parasitica*,

suggesting its involvement in chestnut blight disease as a secondary pathogen of weakened trees [28]. This raises questions about the range expansion of introduced species connecting to environmental changes such as higher temperatures and to the evolutionary possibilities of fungi.

### 4.1.3. Emerging Pathogens

Climatic and environmental drivers could also be involved in the new outbreaks of fungi recently reported on chestnut species [1]. As an example, an emergence caused by the basidiomycota *Fistulina hepatica* (Schaeff.) has been reported in Northern Spain inducing chestnut red stain on *C. sativa* [29,30]. In addition, *Neofusicoccum parvum* (Pennycook and Samuels) Crous, Slippers and A.J.L. Phillips, an emerging fungal pathogen threatening agricultural and forest ecosystems worldwide, was found in association with chestnuts in Sardinia. The same applies for several diaporthalean fungal species, whose taxonomic family includes *C. parasitica*, that have been described as damaging for chestnut trees and nuts. Among these, *Cytospora castaneae* and *Dendrostoma* spp. have been reported as fungal causal agents of cankers on chestnut trees [31]. However, the *Gnomoniopsis* fungal genus is considered as the one in Diaportales including the higher number of species that have been found on hosts belonging to *Fagaceae*, *Onagraceae*, and *Rosaceae* plant families [32].

Even if most of them have been classified as saprobes, colonizing dead leaves, branches, and twigs, there are also *Gnomoniopsis* species that have been reported as pathogens, occurring as host-specific fungi [33]. Among these, *G. smithogilvyi* L. A. Shuttleworth, E. C. Y. Liew, and D. I. Guest (synonym *Gnomoniopsis castaneae* Tamietti) acquired importance as an emerging pathogen affecting chestnut [34]. In addition, an abnormal increase in rotten chestnuts has been observed both in Europe, New Zealand, and Australia since 2005 [34]. In Italy, France, and Switzerland more than 80% of chestnut nuts were affected, causing huge economic losses for the chestnut industry [35]. The phenomenon was firstly addressed to climate-related increases in long-known fungi observed on chestnuts (e.g., *Phomopsis*), and only in 2012 it was described as a new disease [36]. As the pathogen was reported both in Australia and Italy at the same time, it was firstly described as two different species (respectively, *G. smithogilvyi* and *G. castaneae*), and only subsequent studies based on comparative morphology and phylogenesis have demonstrated that the two descriptions were referred to the same organism [4]. Symptoms can be observed on nuts and can be classified as a mummification process of the fruit. As a consequence, the kernel turns brown as soon as the mycelium attacks, causing brown lesions in the endosperm and embryo [37].

*Gnomoniopsis smithogilvyi* was related also to the development of severe cankers on chestnut stem and branches that, in Switzerland, were able to kill up to the 40% of 3-year-old analyzed chestnut trees [31,36,38,39]. Even if phylogenesis and population studies have been conducted concerning *G. smithogilvyi*, its origin and taxonomical features are still not clear, causing debate about the possible existence of different lineages and the legitimate name of the species [40]. However, as ecological interactions among pests and pathogens have been recognized as a possible driver of both fungus and insect spread, some authors have hypothesized that the *G. smithogilvyi* outbreak could be associated with the invasion of insects affecting chestnut [41]. In Italy, the *G. smithogilvyi* outbreak spatially and temporally overlapped exotic pest invasion; no experimental evidence of their connection was found, raising the necessity of deeper understanding the etiology and epidemiology of this fungus, especially concerning its endophytic behaviour [37,42]. A role in *G. smithogilvyi* disease development could also be played by the different fungal communities that can be found as endophytes in different chestnut tissues [43]. As a result, many plant pathogens (e.g., *Colletotrichum acutatum*, *Epicoccum nigrum*, *Stromatoseptoria castaneicola*, *Ramularia endophylla*, *Beauveria bassiana*) were reported as endophytes of chestnut tissues associated with *G. smithogilvyi*. Recently, different *Gnomoniopsis* species have also been reported as pathogens on *Castanea* species resembling the pathogenic behavior of *G. smithogilvyi*. Among these, *Gnomoniopsis daii* C.M. Tian and N. Jiang and *Gnomoniopsis chinensis* C.M. Tian and N. Jiang have been described affecting Chinese chestnuts [32,33].

### 4.1.4. Pests Affecting Chestnut Tree: The Asian Gall Wasp *Dryocosmus kuriphilus*

The Asian gall wasp *Dryocosmus kuriphilus* Yasumatsu is considered worldwide as the most serious pest of the chestnut tree. This insect manipulates the development of the plant inducing the formation of galls on shoots, inhibiting its growth. This action affects flowering and fruiting and limits tree productivity with implications also in wood production. In persisting invasions, the tree experiences a strong decrease in its regeneration capacity due to the reduction in the number of dormant buds [44]. *Dryocosmus kuriphilus* can reduce fruit yield by up to 80%, with important economic losses for chestnut growers. The chestnut gall wasp is native to China but has rapidly spread through Asia, North America, and Europe, where it was reported for the first time in Italy in 2002. The occurrence of *D. kuriphilus* in Europe is likely due to a single introduction of infested chestnuts from China. The colonizing capacity of this wasp is further increased by its reproduction strategy, since females reproduce parthenogenetically with thelytoky. Thus, populations are composed entirely of females, and a single adult can alone establish a new population, laying more than 100 eggs during their ten days of life.

During summer, eggs are located in buds, and after approximately one month the hatching occurs. Larvae form a gall inside which they overwinter until the following spring. New adults usually emerge from June to August; however, a strong influence exerted mainly by temperature on population dynamics and flight phenology has been recently highlighted, with warmer temperatures increasing the survival of immature stages and advancing flight activity [45]. As a result, cold treatment resulted in a reduction in adult emergence and also affected its parasitoid communities which contribute to cynipid mortality [46,47]. Possible interactions between invasive fungal pathogens and pests have been investigated to assess if abandoned galls could act as a fungus entry point. Gall and surrounding tissues have been found to harbor different endophytic communities, suggesting that galls provide an optimal habitat for microorganisms [43]. Moreover, Vannini et al. (2017) [41] assessed that *G. smithogilvyi* strongly affects *D. kuriphilus* mortality due to its pathogenic activity in the galls. According to these authors, *G. smithogilvyi* could be an effective biocontrol agent against these insects but its harmfulness excludes its use in biological control.

### 4.2. Strategies for Prevention, Control, and Management of Pests and Pathologies

An effective strategy to prevent or mitigate the impact of plant pests and pathogens usually rely on early warning and rapid responses [48]. Hence, the possibility to choose among different kinds of actions, ranging from surveillance to removal of diseased trees, is an important part of pest and disease control that allows managers to decide how to allocate funds among different sites. Different strategies against chestnut pests and pathogens have been investigated in the literature [15,49,50]; here following, the most interesting strategies were summarized.

### 4.2.1. Early Detection as Prevention Strategy

The possibility to assess the presence and to quantify the amount of a pathogen even before that the symptomatology can be observed, is a key point for the implementation of efficient prevention, conservation, and management measures [49]. Conventional methods ranging from visual analysis of symptoms to microbiological culturing and biochemical tests can be used for ink disease, *Gnomoniopsis* spp., and *C. parasitica* detection [21,40]. To overcome the problems of these methods, alternative techniques based on immunology (e.g., *Phytophthora* dipstick lateral flow devices) have been implemented [19]. Although these rapid methods show several advantages, they often lack in specificity, making difficult the diversification among taxonomically nearest species [49].

The advent of nucleic-acid-based approaches as polymerase chain reaction (PCR) and its variants (e.g., quantitative real-time PCR) provided sensitive, specific, accurate, and reliable analyses for chestnut diseases [24,37,51]. Moving forward, the detection challenge of plant pathology has become the implementation of point-of-care features of each method

without losing sensitivity, specificity, and accuracy. Hence, new methods ranging from isothermal amplification reactions (e.g., loop-mediated isothermal amplification) to Lab-on-a-chip technologies and biosensors were developed [4]. Moreover, spectroscopic and imaging-based techniques can also be exploited for plant health evaluation. Imaging-based techniques can also be used coupled with mathematical and geostatistical analysis, exploiting predictive models to understand pathogens' capacity of spread and establishment in analyzed areas [52].

### 4.2.2. Control Strategies: Resistance Breeding and Biocontrol

The main control measures applied against chestnut pathogens can be classified as breeding for resistance and biocontrol strategies [48]. The implementation of resistant chestnut trees aimed at reducing the impact of *C. parasitica* in new plantations have been researched since 1983 [22,52]. All these trials led to *C. dentata* × *C. mollissima*, including backcrossing and intercrossing resistance genes against chestnut blight that were developed, analyzed for phenotype, and used in forest restoration campaigns and as rootstock in orchards. However, even if many resistant hybrids *C. crenata* × *C. sativa* are currently on the market, most of the resistance work carried out on *Phytophthora* lacks a direct link among the resistant phenotype and expressed genes, being ascertained only with the advent of new genetic techniques. Among these, CRISPR-Cas9 has become one of the most promising tools for genome editing engineering also usable for controlling plant pathogenic fungi [53].

The use of living organisms able to counteract the action of pathogens is usually preferred as biocontrol. In addition to the use of hypoviruses against *C. parasitica*, many fungal and bacterial strains have been tested both in lab and field trials to counteract the attack of pathogens affecting chestnut trees (e.g., *Trichoderma* spp., *Glomus* spp., *Bacillus subtilis*, *Neopestalotiopsis* sp., *Pestalotiopsis* spp.) [19,21,23,26,39]. Even if this method seems to be promising, it is rarely adopted for forest diseases which have long research tests and are regulated by precautionary principles [50]. Native parasitoids may play an important role in biological control programs, since they can support or reduce the effectiveness of the alien introduced parasitoids. The community of native parasitoids recruited by *D. kuriphilus* has been largely monitored in different countries [54,55]. Surveys highlighted that native parasitoids, especially those associated with oak gall wasps [56], promptly adapted to the new host. Being the autochthonous parasitoids able to shift to *D. kuriphilus*, they represent an important contribution to its regulation [55]. However, their parasitism rates can be variable over time, probably due to the asynchrony between their emergence and the cynipid gall susceptibility; therefore, their action seems to be insufficient to completely control the pests. For this reason, the natural enemy *Torymus sinensis* Kamijo was released in several countries with satisfying results [57].

The introduction of *T. sinensis* has been found to influence native parasitoid richness [54]. Being perfectly adapted to *D. kuriphilus*, it easily establishes itself in the infested territories, monopolizing the pest and bringing native parasitoids back to attack almost exclusively the native oak cynipids [58]. Although biological control is considered the most effective method against *D. kuriphilus*, the dimension and composition of the orchard are important variables. As a result, pest control is more difficult in the case of extensive fields. The use of resistant or less-susceptible cultivars and hybrid clones has been suggested as soon as *D. kuriphilus* proliferated. In Japan, successful resistant breeding of *C. crenata* restrained the infestation for about 20 years, but a novel virulent strain of insect pest overcame plant resistance. In Europe several tests have been performed to assess varietal susceptibility and identify resistant cultivars belonging to the European sweet chestnut. A wide range of variation was observed, but except for the Italian cultivar *"Pugnenga"* and the French cultivar *"Savoye"* [59], the resistant cultivars were from other chestnut species. Only the Italian *"Red Salernitan"* ecotype of *C. sativa* seemed to be resistant, but at a moderate level [60]. Otherwise, *"Bouche de Betizac"* (hybrid *C. crenata* × *C. sativa*) turned out to be efficiently resistant to the insect [57].

Recently, Castedo-Dorado et al. (2021) [61] evaluated the susceptibility to gall wasp of the three chestnut species (*C. sativa*, *C. crenata*, *C. mollissima*) and 27 hybrid clones, finding 2 resistant hybrid clones and 1 hybrid clone with low values of infestation index. They suggested that the cynipid infestation may depend on tree attributes, as genotype and size, especially the height. Overall, the combination between biological control with *T. sinensis* and the use of resistant cultivars or hybrids should be encouraged to limit long-term attacks of *D. kuriphilus*. The application of entomopathogens against the Asian gall wasp was recently experienced by Şahin et al. (2020) [62]. They applied two strains of *Steinernema feltiae* and one strain of *Heterorhabditis bacteriophora* with four dosages against cynipid adults, evaluating their mortality and the number of eggs in buds. The results highlighted that both the entomopathogens were able to infect the cynipid, with a decrease in the average number of laid eggs while increasing dosages. In particular, at 200 IJs/cm$^2$, the mean number of eggs in the treated buds was statistically significant and more than 80% mortality of adults was recorded. This experiment gave insight into the possibility to apply this environmentally friendly approach.

4.2.3. Management Strategies

Different silvicultural treatments can be applied to stimulate host resilience and to contrast chestnut diseases. Management with grafting, pruning, and wounding could be valuable against *C. parasitica*, but it could reduce the effect of hypoviruses when choosing biocontrol [23]. Moreover, leaving organic residues in orchards can reduce soil fertility and facilitate the spread of some fungi (e.g., *Gnomoniopsis* spp.) [42]. Concerning *Phytophthora* management, silvicultural treatments can be a good alternative to give vigor to infected plants, but they need to be coupled with drainage interventions in order to stop the ability of *Phytophthora* to move and spread [19]. Alternatively, chemical application has been explored; however, to overcome their environmental effects, chemical application by trunk injections has also been researched as an endotherapy treatment [16].

Since *T. sinensis* requires some years to reduce *D. kuriphilus* incidence to acceptable levels, quicker strategies were investigated. Maltoni et al. (2012) [63] tested the effect of green pruning since this technique stimulates the formation of new healthy shoots, potentially postponing the production of buds long enough to limit *D. kuriphilus* oviposition. Authors applied two different types of shortening consisting of eliminating up to 1/3 or 2/3 of shoot length and nodes, and tested four timings between the second half of May and the second half of August. Results showed that pruning was effective in reducing *D. kuriphilus* damage but the combination between pruning type and application time determined different responses on development and phytosanitary status, with opposite trend during the growing season. Pruning could be a useful method in combination with biological control; however, this technique seems to be laborious and expensive for large commercial growers [57]. Fernandez-Conradi et al. (2018) [64] found that chestnut–oak and chestnut–ash mixtures were significantly less infested than pure chestnut stands and chestnut–pine mixtures, suggesting that the quantity of non-host trees in mixed stands strengthens the associational resistance to invasive pests. Ciordia et al. (2020) [65] applied hot water baths to dormant chestnut buds in order to disinfect them from *D. kuriphilus*. By soaking dormant scions at 49 °C for 10 min, authors found an effective result in killing insect larvae while retaining a high percentage of successful grafts.

## 5. Chestnuts
### 5.1. Main Criticalities in Harvest, Post-Harvest, Processing, and Storage
5.1.1. Insects and Their Control Strategies

Chestnuts are rich in starch, moisture, and sugars; these characteristics make them particularly suitable to insect attacks, with negative consequences on the yield and on the quality of the harvest [3]. *Curculio elephas* Gyll. and three tortricid moths, namely, *Pammene fasciana* L., Cydia *fagiglandana* Zel., and *Cydia splendana* Hb, represent the main insects that can attack *C. sativa*. These insects may cause severe damage with infestation

starting when fruits are still on trees. *Curculio elephas*, known as the chestnut weevil, is recognized as the major pest of European sweet chestnuts, causing up to 40% fruit loss. This species is univoltine. Adults emerge from the soil in summer and start to reproduce, feeding on the tree buds. Females lay eggs on or inside chestnuts, creating holes thanks to the rostrum. One or two eggs occur approximately per single fruit. For two months, larvae feed on the kernel, then leave the fruit and burrow into the ground to overwinter. The pupal period lasts on average 9 months [66,67]. Weather conditions and chestnut quality mainly influence female fecundity, adult emergence, and individual survival [66].

*Pammene fasciana*, known as the early chestnut moth (since it flies from June to July), is considered as a minor pest in many European countries since it occurs at the beginning of the development of chestnuts, resulting in an early drop in the attacked fruits. Conversely, *C. fagiglandana*, also called the intermediate chestnut moth (flying from the end of July to September), and *C. splendana*, the late chestnut moth (which occurs in September), received more attention because their attacks may incur more critical consequences. Damages are caused by the endophytic activity of the larvae which feed on chestnuts for about a month. These species are univoltine and overwinter as mature larvae into the grooves of the tree bark or on the ground. Females lay hundreds of eggs on the underside of leaves, close to the frutescence. The larvae penetrate the pericarp and dig tunnels into the fruits. It has been observed that *C. splendana* attacks most often the fruits smaller in size, probably because the hilum of bigger chestnuts is tougher, meaning it is harder to perforate [67].

Insect control in pre-harvest could be performed using entomopathogenic fungi (EPF) and nematodes (EPN) against the chestnut weevil and tortricid moths. The nematode *Heterorhabditis bacteriophora* was effective against chestnut weevil larvae. It was tested alone or in combination with the EPN *Steinernema glaseri* and *S. weiseri*; however, higher larval mortality was not achieved in co-application than with one species treatment [68]. Asan et al. (2017) [69] performed several experiments to evaluate the capacity of the fungus *Metarhizium brunneum* and the nematode *H. bacteriophora*, alone or in combination, to control *C. elaphus* and tortricid moths. They found that the EPF was overall more virulent than the tested EPN, although the latter killed chestnut weevil larvae quickly. The combination of the two entomopathogens does not provided supplementary advantages over the use of *M. brunneum* alone, except for an earlier mortality of larvae which was not sufficient to justify the application of both fungus and nematode. *Metarhizium brunneum* caused a higher mortality compared with *S. feltiae*, but with the co-application of the two, 100% success in controlling this insect was achieved, since their interaction was additive. The results of the latter authors are supported by other authors in the literature [70].

In post-harvest control of insects, HPP, Ohmic heating, and radio frequency treatments seem to be the most interesting innovative applications. Radio frequency treatment, used alternatively to conventional thermal techniques, could be effectively used as a physical method to disinfect agricultural commodities and to protect stored products from insects, including *C. elaphus*. Hou et al. (2015) [71] determined the suitable radio frequency treatment protocol to control chestnut weevils, ascertaining that heating around 50 °C with a holding time of 2 min caused 95.65% of larvae mortality, and, additionally, limited moisture losses in fruits during the storage, keeping unchanged texture characteristics and quality parameters. Another emerging post-harvest technology is Ohmic Heating (OH). Pino-Hernández et al. (2021) [72] evaluated the possibility to apply this technology in the chestnut industry. They compared pest infestations, physicochemical, and nutritional properties of fruits treated with OH at different temperatures with other products untreated or subjected to conventional hydrothermal processing. The results of the latter authors showed that the best treatment to control molds and insects was OH conducted at 55 °C followed by a storage at 5 °C and 70% RH. In these conditions, chestnut shelf-life increased and physiochemical quality remained substantially unchanged. Last but not least, HPP processing might represent another alternative nonthermal technology to preserve different products against pathogens without compromising the quality. Pino-Hernández et al. (2022) [73] evaluated this approach, treating chestnuts at different pressures (400, 500, and

600 MPa for 5 min at 20 °C) and estimating fungus or insect infestations, physiochemical and nutritional characteristics of fruits, as well as the shelf-life 60 days after the treatment. HPP proved to be able to eliminate pathogens without affecting ascorbic acid content. Moreover, a softening and easy peeling of treated fruits were found.

5.1.2. Molds and Toxins

Chestnuts can be contaminated by molds not only before their harvest. In particular, this contamination can also occur during transportation, storage, and processing. Several mycotoxin-producing fungi, such as *Fusarium* spp., *Penicillium* spp. and *Aspergillus* spp., have been isolated from chestnuts [7]. As a result, several mycotoxin contaminations on chestnuts and derived commercial products, have been reported in different countries. Prencipe et al. (2018) [7], carried out an important monitoring process on fresh chestnuts from orchards, on dried chestnuts, chestnut granulates, and chestnut flour with the aim of identifying the species of *Penicillium* through molecular and macromorphological analyses. The results of Prencipe et al. (2018) [7] identified twenty species, divided into 2 subgenera (*Aspergilloides* and *Penicillium*) and 8 sections (*Aspergilloides*, *Brevicompacta*, *Chrysogena*, *Citrina*, *Exicauilis*, *Fasciculata*, *Penicillium* and *Robsamsonia*). According to the source of isolation, 8 species were found by the latter authors from the orchard samplings, 12 species from the chestnut processing, 9 species from the flour, and 8 species from the indoor samplings. *P. crustosum, P. glabrum* and *P. bialowiezense* were the predominant species, and they represented 57% of the isolates [7]. *P. crustosum* was the predominant species, with 44 isolates from 124 total; this confirm that *P. crustosum* is a ubiquitous species able to survive under different environmental conditions [7].

With respect to mycotoxin production, the study of Prencipe et al. (2018) [7] highlighted that 59% of the analyzed strains (41/70) were able to produce at least one mycotoxin on chestnuts. In conclusion, according to the findings of Prencipe et al. (2018) [7], the isolation of different *Penicillium* species from all the investigated samples and their mycotoxin production are causes of concern and a significant alarm bell due to their effects on human health. The results were supported by Pietri et al. (2012) [74]. The latter authors collected and analyzed thirty-seven samples of chestnut flour and fourteen of dried chestnuts from retail outlets located in the north of Italy. Pietri et al. (2012) [74] highlighted that the mycotoxin contamination was widespread and remarkable. In particular, with respect to aflatoxins, the incidence of aflatoxin B1 was 62.2% and 21.4% in chestnut flour and dried chestnuts, respectively. In the same products, the percentage of samples exceeding the value of 2.0 mg per Kg for aflatoxin B1 (maximum limit fixed by EC Regulation 165/2010 in dried fruits) was 24.3% and 7.1%, respectively [74]. Moreover, Pietri et al. (2012) [74], found that the maximum values for aflatoxin B1 and total aflatoxins were 67.88 and 188.78 mg per Kg for chestnut flour and dried chestnut. Ochratoxin A occurred in all samples, showing very high values with a mean value of 12.38 and 13.63 mg per kg for chestnut flour and dried chestnuts [74]. The percentages of samples exceeding the limit of 10 mg/kg were 64.9% and 42.8% for chestnut flour and dried chestnuts, respectively, and the maximum value was 65.84 mg/kg in dried chestnut sample [74].

Bertuzzi et al. (2015) [75] support these findings. In particular, Bertuzzi et al. (2015) [75] highlighted that, in fresh chestnuts, mycotoxins were rarely detected, whereas a widespread contamination was found in dried products, particularly in chestnut flour. The incidence of aflatoxin B1 was 92.0% and 40.0% in chestnut flour and dried chestnuts, respectively [75]. Moreover, in chestnut flour, the percentage of samples exceeding the law limit value of 2.0 mg kg$^{-1}$ for aflatoxin B1 was 24.0% [75]. In addition, chestnut flour was also often contaminated with ochratoxin A, citrinin, roquefortine C, and mycophenolic acid [75]. These results clearly highlight that chestnut and chestnut flour contamination mainly occurs in post-harvest, drying, storage, and sorting.

## 5.2. Challenges and Opportunities in Chestnut Drying

Chestnut drying is a key unit operation for the safety and the quality of chestnut flour. To obtain flour, chestnuts are dried, and, successively, the pericarp and endocarp are removed before milling. Nowadays, both traditional and industrial methods are used [76]. In the first, the drying of chestnut is carried out using a traditional dry kiln, which usually is building with a two-floor cabin built with local stone [76]. On the ground floor, heat is produced by fire made with wood and scraps of chestnut tree [76]. On the first floor, homogeneous layers of chestnuts are laid on a rack for drying [76]. Generally, this process is performed at 40 °C for 30 days and chestnuts are turned occasionally to ensure uniform drying [76]. On the contrary, industrial processes use modern drying and milling systems. In particular, chestnut drying is performed in modern kilns at higher temperature (up to 70 °C or higher) for shorter period of time (from hours to a few days). Clearly, the use of different approach, drying temperature, and drying systems has distinct effects on chestnut flour safety and quality.

Correia et al. (2009) [77] assessed the effects of drying conditions on the morphological and chemical properties of chestnuts (*Longal* and *Martainha* varieties). The results of the latter authors found that chemical composition of flour, morphological properties of starch, and all drying curves were different according to drying temperatures (from 40 °C to 70 °C). In particular, the higher the drying temperature, the higher the reducing sugar content and the lower the starch content [77]. However, *Martainha* and *Longal* proved to be differently affected by drying conditions. Correia et al. (2009) [77] found that *Longal* presented bigger starch granules, whiter flours, higher reducing sugar content, and lower starch and sucrose contents. *Martainha* flours, instead, showed less starch damage [77]. Therefore, it could be very interesting to investigate and develop specific drying schedules optimized according to the processed chestnut cultivar.

Another interesting innovation in chestnut drying is proposed by Ahmed and Al-Attar (2015) [78]. The latter authors tested the effects of freeze-drying (FD) and tray-drying (TD) process on the functional, thermal, and rheological properties of chestnut flour doughs. The results of Ahmed and Al-Attar (2015) [78] highlighted that the drying method did not influence the physicochemical properties of the flour but influenced significantly particle mass distribution and color values. The thermal analysis of chestnut doughs showed two distinct peaks related to starch gelatinization and melting of starch–lipid complexes, respectively [78]. Moreover, the amylograms of the chestnut dispersions showed higher maximum viscosity and heat stability for the FD sample [78]. In addition, the average particle size of the tray-dried sample was larger than the freeze-dried sample [78]. In conclusion, although the freeze-dried chestnut flour maintained the microstructure in the dough and showed optimal viscoelastic and pasting properties compared with the tray-dried sample, the applicability of this drying method to an industrial scale must be carefully evaluated.

## 6. Chestnut Flour and Applications in the Food Industry

### 6.1. Effects of Drying Conditions on the Safety and Characteristics of Flour, Doughs, and Products

In addition to Sections 5.1.1 and 5.1.2, not only pests, molds, and toxins could represent a critical issue during storage [79]. Other pathogens (e.g., *Salmonella*) can be a critical source of contamination. Sharma et al. (2021) [80], determined the survival kinetics of *Salmonella* in chestnut flours at three different RH (i.e., 25%, 45%, and 70%) during a 120-day storage period. The results of the latter authors revealed that the storage RH significantly affected the growth kinetics of *Salmonella* which showed improved survival kinetics at low RH (25% or 45%) compared with high RH conditions (70%), supporting other literature reports on the persistence of *Salmonella* at low $a_w$ conditions for extended periods. In addition, Sharma et al. (2021) [80] found that *Salmonella* survival improved in chestnut flour due to a higher availability of nutrients. In conclusion, given the high risks linked to *Salmonella*, the prevention of the risk of contamination appears essential in order to avoid hazards for consumer health.

With respect to the effects of chestnut drying on dough rheological properties, Moreira et al. (2013) [81] assessed the effects of three different chestnut drying temperatures (45, 65, and 85 °C) on the rheological properties of doughs produced using chestnut flour. The results of the latter authors highlighted that in oscillatory (1–100 rad s$^{-1}$ at 0.1% strain), temperature sweep (30–90 °C), and creep recovery (loading 50 Pa for 60 s) tests, the doughs obtained from chestnuts dried at 85 °C showed the most interesting properties (in particular, a remarkable elasticity associated with starch gelatinization). More precisely, Moreira et al. (2013) [81] found that drying the chestnuts above the onset temperature of chestnut starch gelatinization significantly modifies the most important properties of doughs. As a result, the doughs obtained from chestnuts dried at 85 °C had higher water absorption (WA) and elasticity.

Additionally, Correia and Beirão-da-Costa, (2012) [82] tested the effects of different drying temperatures (40, 50, and 60 °C) on starch-related functional and thermal properties of chestnut flours. According to the results obtained by the latter authors, it is possible to conclude that drying temperature has significant effects on starch-related chestnut functional properties. In particular Correia and Beirão-da-Costa, (2012) [82] found that starch suspensions from fruits dried at 60 °C showed higher viscosity due to higher values of amylose and resistant starch content. Moreover, Correia and Beirão-da-Costa (2012) [82] highlighted a decrease in transition temperatures and in enthalpy with the increase in drying temperatures. Contrarily to Moreira et al. (2013) [81], the latter authors found that the flour obtained by chestnut dried at 60 °C had the best functional properties as a food ingredient [82]. In conclusion, it seems that further investigations are necessary to find the optimal drying temperature for chestnut flour doughs and products, with the additional problem related to cultivars variability.

Finally, Wani et al. (2017) [83] compared the effects of pan and microwave roasting on physicochemical, functional, rheological, and antioxidant properties of chestnut and chestnut flour. The latter authors did not find differences between the two roasting methods and the native chestnut with respect to protein, fat, and ash contents. On the contrary, "L" values decreased from 90.66 to 81.43, whereas "a" and "b" values increased from 0.02 to 0.90 and 11.99 to 20.5, respectively, upon roasting [83]. Moreover, Wani et al. (2017) [83] found a significant increase in water absorption (1.32–3.39 g/g), oil absorption capacity (1.22–1.63 g/g), and antioxidant properties in the case of roasting. Flour obtained from pan-roasted chestnuts exhibited a significant decrease in light transmittance, foaming, and pasting properties, whereas higher gelatinization temperatures and lower enthalpies were reported in microwave roasted chestnut flours [83]. Finally, roasted flours had higher total phenolic content and antioxidant activity compared with their native counterparts. However, microwave-roasted chestnut flours had superior nutritional quality in terms of their antioxidant potential compared with pan-roasted chestnut flours [83].

*6.2. Innovative Applications of Chestnut Flour in the Food Industry*

Chestnut flour has been proposed as a very interesting ingredient for the production of innovative and fortified product formulations [84,85]. The literature review highlighted several applications of chestnut flour in the food industry. According to the results, the production of bread, pasta, and bakery products seems to be the main applications. In particular, given the absence of gluten and the excellent nutritional and organoleptic properties, chestnut flour has been frequently investigated for use in the development of gluten-free products. The following authors explored the use of chestnut flour in different research fields of the food industry.

Sirini et al. (2020) [9], investigated the effects of chestnut flour incorporation into Longaniza de Pascua (a traditional dry-cured meat sausages produced in the Valencian community (Spain)). The latter authors found that the addition of chestnut flour reduced the presence of residual nitrite and increased the dietary fiber and polyphenol content. Moreover, the chestnut flour seems to be a very promising prebiotic for the cured meat industry [9]. Another interesting application is suggested by Ozcan et al. (2017) [86]. The

latter authors assessed the effects of chestnut flour in stimulating the growth of probiotic bacteria in a fermented skim milk produced with different probiotic strains (*Lactobacillus acidophilus*, *L. rhamnosus* and *Bifidobacterium animalis subsp. Lactis*). The results of Ozcan et al. (2017) [86] highlighted that all probiotic fermented milks enriched with chestnut flour displayed significant probiotic viability (>7 log10 cfu/g) with high antioxidant capacities. Moreover, the antioxidative potential of probiotic fermented milks clearly highlighted how chestnut flour could be effectively incorporated in different dairy foods.

Littardi et al. (2020) [84], investigated the effect of different levels of chestnut flour in formulations of soft wheat fresh pasta. The substitution with chestnut flour increased the water absorption properties compared with the control [84]. At the molecular level, Littardi et al. (2020) [84] highlighted different levels of water redistribution in pasta containing chestnut flour that was related to worse macroscopic quality properties. With respect to the cooking phase, chestnut enriched pasta showed higher solid loss for increasing levels of substitution [84]. Moreover, a significant reduction in hardness and deformability was found by the latter authors. On the other hand, increasing percentages of chestnut flour increased the darkening of the final products and the antioxidant activity (even after cooking) [84]. Another interesting application is suggested by Alinovi et al. (2022) [87], which tested the effect of substituting 3 g of wheat flour with an equivalent amount of fiber rich ingredients named chestnut peels (CP) and wheat bran (WB) in the breadmaking process. With respect to CP incorporation, the results of Alinovi et al. (2022) [87] highlighted a significant effect on the physicochemical characteristics of the product (some of them critical for the consumer appreciation of a bread). Nevertheless, despite the necessity of further studies, the incorporation of chestnut peels might be interesting for the production of fiber-enriched products that can contribute to the development of a more sustainable and efficient chestnut production chain [87].

Chestnut flour does not contain gluten. For this reason, chestnut flour is suitable for the production of bakery products for people with celiac disease or gluten sensitivity. With respect to gluten-free applications, Rinaldi et al. (2017) [88] investigated the effects of sourdough fermentation combined with chestnut flour incorporation in the improvement of the nutritional quality of gluten-free bread during a 5-day shelf life. Chestnut flour limited the acidification of both dough and breads reducing the decrease in water-holding capacity and the increase in crumb firmness [88]. Although both sourdough and chestnut flour addition led to a reduction in loaf volume (as confirmed by Venturi et al. (2022) [89]), a significant reduction in the staling phenomenon at 5 days of storage has been observed by Rinaldi et al. (2017) [88]. In conclusion, the combination between sourdough fermentation and chestnut flour addition could improve gluten-free bread characteristics, even if further improvements need to be found to increase the gluten-free bread volume.

Paciulli et al. (2018) [10] tested the effect of different levels of chestnut flour substitution (0, 500, 800, 1000 g/kg) in gluten-free biscuits' formulations during a storage time of 60 days. The results found that chestnut flour increased water binding capacity and decreased water absorption of the gluten-free mix used [10]. During storage, all the formulations showed an increase in water content from 26 to 35 g/kg [10]. Contemporaneously, hardness values also increased during storage (with the exception of samples produced with 1000 g/kg chestnut flour) [10]. In addition, gluten-free biscuits produced with chestnut flour appeared darker (lower L* and higher a* and b* values) and with higher oxidative stability values, probably for the polyphenol content of chestnut flour [10]. In conclusion, Paciulli et al. (2018) [10] highlighted that chestnut flour improved the technological and organoleptic quality of the gluten-free biscuits. In particular, the latter authors reported that a substitution with 500 g/kg of chestnut flour might represent the best compromise between quality and storage stability.

Also, Torra et al. (2021) [90] and Silav-Tuzlu and Tacer-Caba (2021) [91] tested the addition of chestnut flour in the production of gluten-free biscuits with similar results. According to Torra et al. (2021) [90], the addition of chestnut flour to the biscuit formulation increased the values of G' and G'' but reduced the loss factor compared with the doughs

made using chickpea flour. Moreover, the substitution with chestnut flour significantly decreased the diameter and the spread ratio of the cookies, while increasing the hardness and darkening of the final product [90]. These results are supported by Silav-Tuzlu and Tacer-Caba (2021) [91]. In particular, according to the results of the latter authors, the gluten-free biscuits formulated with chestnut flour had the highest phenolic content (400.2 mg GAE per 100 g dry sample) and the highest total antioxidant activity (155.5 mg Trolox equivalent (TE) per 100 g dry sample). Moreover, the gluten-free biscuits produced using chestnut flour were not affected by oxidation, contrary to gluten-free biscuits formulated with chia seed flour [91]. In conclusion, the incorporation of chestnut flour in gluten-free biscuits and in other bakery products seems to be able to remarkably improve the total dietary fiber content, the nutritional characteristics, and, finally, the sensory profile of the product.

*6.3. Valorization of Chestnut by-Products*

The recovery and valorization of nutraceutical components from bio-wastes produced in different agro-food production chains allow to face the monumental challenge of environmental sustainability, with additional economic benefits. With respect to chestnut production chain, chestnut shells represent one of the most interesting by-products. The literature review found that total phenolic content in chestnut (*C. sativa*) shell extracts varied between 26.2 and 59.7 g GAE/100 g extract. Ham et al. (2015) [92], investigated different methods for phenolic compounds extraction from chestnut inner shell. In particular, the latter authors tested aqueous alcohols and alkaline solutions (50% ethanol, 50% methanol, 1% NaOH, and 2% NaOH) at different temperatures (25–90 °C). The results highlighted that total phenolic content and antioxidant activity were increased as the extraction temperature increased [92]. However, the phenolic composition and antioxidant activity were significantly different among the extracts prepared under different conditions [92]. Aqueous ethanol (50%, $v/w$) was most effective in extracting the total phenolics, resulting in the highest DPPH radical scavenging activity [92]. The alkaline solution instead, appeared more effective in extracting specific phenolics, such as tannins and flavonoids [92]. In conclusion, the results obtained by Ham et al. (2015) [92] found that chestnut inner shell might be effectively used as a source of natural antioxidants and deodorants in the food industry, but avoiding excessive heating (>70 °C).

Additionally, Squillaci et al. (2018) [93] investigated the recovery of polyphenols from outer chestnut shells (IOCS), and inner chestnut shells (ICS) through an eco-friendly method. The results of the latter authors highlighted that IOCS extract contained the highest number of polyphenols (205.99 $\pm$ 13.10 mg of Gallic Acid Equivalents/g of dry extract). In addition, Squillaci et al. (2018) [93] found that condensed tannins represented the main phenolic fraction (78.88% and 59.14% of the total phenolic compounds in IOCS and ICS extracts, respectively). The results of the latter authors are supported by Pinto et al. (2021) [94], which highlighted that several bioactive compounds such as ellagitannins, condensed tannins, phenols, acids, and flavonoids can be extracted from chestnut shells. In particular, Pinto et al. (2021) [94], highlighted that tannins are the predominant polyphenolic class (approximately 60% of active substances including castalagin, castalin, vescalagin, and vescalin, which are easily hydrolysable).

Not only wasted shells may be used for the recovery of polyphenols; as highlighted by Vella et al. (2018) [95], it is also possible to recover these bioactive compounds from chestnut tree leaves and burs. The latter authors tested three different extraction methods (i.e., methanol 60%, ethanol 60%, and boiling water). The results of Vella et al. (2018) [95] highlighted that boiling water was the best extraction solvent with respect to polyphenol recovery from chestnut shells and burs. On the other hand, 60% ethanol proved to be the most efficient for leaves. In particular, the highest polyphenol contents were 90.35, 60.01 and 17.68 mg gallic acid equivalents g$^{-1}$ in leaves, burs, and shells, respectively. According to the results of Vella et al. (2018) [95], it is possible to recovery polyphenols from different wastes of chestnut production chain using an ecofriendly extraction process.

Additionally, Munekata et al. (2016) [96] tested the extraction of polyphenols from chestnut tree leaves with satisfactory results. In particular, the latter authors highlighted that a total of 15 compounds corresponding to flavonoids (quercetin and cirsiliol), phenolic acids (gallic, protocatechuic and vanillic acids) and lignans (medioresinol), were identified in chestnut tree leaf extract [96].

Recently, an interesting approach for the production of ellagic acid from wasted chestnut shell (inner pellicle and peel) by fungal fermentation (black *Aspergillus* spp. (*A. aculeatus* ZGM6, *A. japonicus* ZGM4, *A. niger* ZDM2, *A. tubingensis* ZDM1)) has been proposed by Gulsunoglu-Konuskan et al. (2021) [97]. The results of the latter authors highlighted that *A. japonicus* ZGM4 was found to yield the highest amount of ellagic acid. Fermentation of chestnut shell under the optimized conditions (11.5 g/L lactose, 5.9 g/L yeast extract, 140 h) generated a six-fold increase in the concentration of ellagic acid [97]. Despite further investigations are needed, the new bioprocess for the production of ellagic acid suggested by Gulsunoglu-Konuskan et al. (2021) [97] is very interesting since it valorizes food wastes for the production of bioactive compounds, with economic and environmental advantages compared with other methods.

Finally, another interesting strategy for the valorization of chestnut shells is suggested by Gullón et al. (2018) [98]. The latter author tested hydrothermal treatment of chestnut shells for the solubilization of hemicellulosic oligosaccharides and antioxidant compounds. Gullón et al. (2018) [98] found that operational conditions of 180 °C allowed to obtain 18.3 g oligosaccharides/L, with a rich substitution pattern and limited formation of degradation products (0.51 g/L of acetic acid). In addition, the autohydrolysis liquors had good phenolic compounds content. In conclusion, despite autohydrolysis products might need further fractioning and refining processes, the findings of Gullón et al. (2018) [98] suggested an alternative process to obtain oligosaccharides and antioxidant compounds which may be used as functional ingredients in food industry.

## 7. Conclusions and Future Trends

Given the fundamental role of chestnut production chains in several industries (timber, flour, honey, and tannins) and considering the need to face the monumental challenge of environmental sustainability, a relaunch of this production chain is definitely essential. This review highlighted the main criticalities related to chestnut tree cultivation, chestnut storage and processing, flour production, and by-product valorization. With respect to chestnut tree cultivation, considering the aggravating factor of climate change, it is essential to develop and extensively apply specific strategies based on early detection and control, in order to contrast known and emerging pests and diseases. This challenge could be faced using an integrated framework of prevention, control, and management strategies. In particular, specific strategies to contrast the emerging pathogen *Gnomoniopsis smithogilvyi* are urgently needed, since it can lead to the total loss of the harvest. In addition, its endophytic behaviour makes the exploitation of effective control measures difficult.

Concerning chestnut storage, this review found that the most critical problems are related to mold development and mycotoxin production, both in chestnuts and chestnut flour. Several mycotoxin-producing fungi, such as *Fusarium* spp., *Penicillium* spp. and *Aspergillus* spp., have been isolated. As a result, various mycotoxin contaminations on chestnuts and derived commercial products have been reported in different countries. To face this problem in post-harvest, HPP, Ohmic heating, and radio frequency treatments seem to be the most interesting innovative applications. The results of this review also demonstrated the effectiveness of these innovative treatments to solve the problem of pest contamination in post-harvest.

In chestnut drying and processing to produce flour, particular attention needs to be paid to mold and mycotoxin contaminations, which definitely represent the main criticalities of chestnut flour. In this direction, only effective prevention strategies are able to control the occurrence of these and other pathogens. In addition to molds and mycotoxins, chestnut drying is the key unit operation able to ensure the safety and the

quality of chestnut flour. For this reason, further investigations are needed to improve the dying process in both traditional and modern dry kilns, with the possibility to investigate and develop specific drying schedules optimized according to the processed chestnut cultivar. Moreover, research efforts should also focus on the development of an innovative drying process for chestnuts, able to improve the safety and the nutritional quality of chestnut flour.

Finally, to fight the challenge of environmental sustainability, the valorization of the whole chestnut by-products (wood from pruning, leaves, burs, and shells) can play a key role in the relaunch of this production chain, also from an economic point of view. This review clearly summarized that the recovery and valorization of nutraceutical components (mainly polyphenols) from chestnut by-products is the most interesting, sustainable, and profitable strategy. However, the fungal fermentation or the incorporation of little amounts of these by-products (mainly chestnut shells) in foods, seem to be interesting alternatives. In conclusion, the application of the innovations proposed in this review could significantly improve chestnut tree cultivation (increasing yields and quality), post-harvest management, and flour production, with an important revitalization of the entire supply chain both in environmental and economic terms.

**Author Contributions:** C.A.: conceptualization, methodology, validation, data curation, writing (original draft), writing (review & editing), visualization. A.C.: conceptualization, methodology, validation, data curation, writing (original draft), writing (review & editing), visualization, project administration. A.A.: conceptualization, methodology, validation, data curation, writing (original draft), writing (re-view & editing), visualization. All authors have read and agreed to the published version of the manuscript.

**Funding:** This work was partially funded by "Fondazione CR Firenze".

**Conflicts of Interest:** The authors declare no conflict of interest.

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
