# Peer review of "From Chestnut Tree (Castanea sativa) to Flour and Foods: A Systematic Review of the Main Criticalities and Control Strategies towards the Relaunch of Chestnut Production Chain"

_sustainability, doi:10.3390/su141912181_

Round 1

Reviewer 1 Report

General comment

 The review summarized the knowledge regarding the main criticalities in chestnut tree cultivation, chestnuts processing, and in chestnut flour production suggesting improvement strategies to (i) contrast the main pests and diseases affecting chestnut tree, (ii) improve chestnut processing and flour production, and, (iii) valorize by-products generated by chestnut production chain.

A systematic review was performed and the articles found were well managed allowing an easy reading. The issues about chestnuts, well-argued by the Authors, are numerous leading to a quality description of issues without quantitative considerations.

The topic shows a sufficient level of novelty and could result interesting for the readers of the Journal.

I find the “Conclusions & future trends” section a bit poor. For this reason I suggest to improve and further discuss this crucial aspect.

Some minor correction and improvement should be provided (see specific comments).

Decision: the manuscript needs to be slightly improved before publication. Minor revisions are proposed.

Specific comments

 Affiliations

Check if they are correct.

Introduction

Line 44 check space at line 44

2. Search strategy

The sentence “Figures 1, 2, and 3 summarize, in the form of flow chart…” Is the same at lines 98-99 and 114-115. Delete one.

Conclusions & future trends

It could be deepened, for example, by highlighting the advantages brought about by the solutions only listed by the authors after analysis of the literature. Indicating the potential benefits for the supply chain.

Author Response

Dear Editor,

the authors would like to thank reviewer 1 for his valuable comments. Here following, you can find a reply point by point to the reviewer suggestions.

Rev 1: General comment

The review summarized the knowledge regarding the main criticalities in chestnut tree cultivation, chestnuts processing, and in chestnut flour production suggesting improvement strategies to (i) contrast the main pests and diseases affecting chestnut tree, (ii) improve chestnut processing and flour production, and, (iii) valorize by-products generated by chestnut production chain.

A systematic review was performed and the articles found were well managed allowing an easy reading.

Answer: thanks. We really appreciate it

Rev 1: The issues about chestnuts, well-argued by the Authors, are numerous leading to a quality description of issues without quantitative considerations.

Answer: thanks. We are glad that the quality of the description of issues was judged as good.

Rev 1: The topic shows a sufficient level of novelty and could result interesting for the readers of the Journal.

Answer: Thanks. We really appreciate this comment.

Rev 1: I find the “Conclusions & future trends” section a bit poor. For this reason I suggest to improve and further discuss this crucial aspect.

Answer: done, thanks. Please see new conclusions section.

Rev 1:  Some minor correction and improvement should be provided (see specific comments).

Answer: done, thanks. All the reviewer 1 comments have been addressed (see point by point answers)

Rev 1: Decision: the manuscript needs to be slightly improved before publication. Minor revisions are proposed.

Answer: Thanks, we really appreciated the minor revisions suggested by reviewer 1. All the suggestions have been addressed.

Rev 1: Specific comments. Affiliations: Check if they are correct.

Answer: thanks for the comment. Affiliation have been corrected.

Rev 1: Introduction. Line 44 check space at line 44

Answer: done, thanks. The space has been removed.

Rev 1:  2. Search strategy

The sentence “Figures 1, 2, and 3 summarize, in the form of flow chart…” Is the same at lines 98-99 and 114-115. Delete one.

Answer: done, thanks. The sentence in the “results of the systematic literature review” section has been changed.

Rev 1: Conclusions & future trends

It could be deepened, for example, by highlighting the advantages brought about by the solutions only listed by the authors after analysis of the literature. Indicating the potential benefits for the supply chain.

Answer: done, thanks. Please see new conclusions section.

Reviewer 2 Report

This is a very interesting review and well supported with the right literature. However, I will suggest that the authors edit the maniscript for english editing purposes as  they are using some wordings which are not scientifically correct.  find below some sugestions....

Line 16: replace work by review and  replace  this "review" by : the first aim of this overview on the literature is to....

Line 35: rephrase the sentence by Chestnut species are of high importance due to....

Line41: For this reason, they have been  cultivated (or maybe planted???) since...

 Line 46: remove " having climatic correspondance wit its Ponto-caucasian origin".

Line 48: remove also and add  it to chestnut can also be...

Line 49: remove " rarely exceeding 1200 m of altitude.

Line 61: add to an UN-FAO report.

Line 72: remove "are a good source of"

LIne 79 to 87: very long 2 sentences and should be rephrased or segmented into more sentences.

Line 199: In emerging pathogens:  what is the risk of cross infections from other  type of trees  near  the the trees?  what about Ceratocystis fimbriata complex ? as it occurs in Italy on platan. Maybe you can separate the root rote pathogens from the others??? 

Maybe  creating a Table specific to chestnut  for each  headings with name of pathogens or process etc... and  linked literature will be useful for the readers. Although, I understand that due to copyrights  regulations, it is difficult to inser some photos, It could be vauable to include some photos which are free of charges in order to illustrate some of the aspects of this review.

Overall, congratulations to the authors as this is an extensive well focussed review.

Author Response

Dear Editor,

the authors would like to thank reviewer 2 for his valuable comments. Here following, you can find a reply point by point to the reviewer suggestions.

Rev 2: This is a very interesting review and well supported with the right literature.

Answer: thanks for this comment. We really appreciate it.

Rev 2: However, I will suggest that the authors edit the manuscript for English editing purposes as  they are using some wordings which are not scientifically correct.  find below some suggestions.... Line 16: replace work by review and  replace  this "review" by : the first aim of this overview on the literature is to....

Answer: done, thanks. Please see line 14.

Rev 2: Line 35: rephrase the sentence by Chestnut species are of high importance due to....

Answer: done, thanks. Please see line 33

Rev 2: Line41: For this reason, they have been  cultivated (or maybe planted???) since...

Answer: done, thanks. Please see line 39.

Rev 2: Line 46: remove " having climatic correspondence with its Ponto-Caucasian origin".

Answer: done, thanks. Please see line 42–44.

Rev 2: Line 48: remove also and add  it to chestnut can also be...

Answer: done, thanks. Please see line 44

Rev 2: Line 49: remove " rarely exceeding 1200 m of altitude.

Answer: done, thanks. Please see line 45

Rev 2: Line 61: add to an UN-FAO report.

Answer: done, thanks. Please see line 59

Rev 2: Line 72: remove "are a good source of"

Answer: done, thanks. Please see line 70.

Rev 2: Line 79 to 87: very long 2 sentences and should be rephrased or segmented into more sentences.

Answer: done, thanks. Please see lines 69–76

Rev 2: Line 199: In emerging pathogens:  what is the risk of cross infections from other  type of trees  near  the trees?  what about Ceratocystis fimbriata complex ? as it occurs in Italy on platan. Maybe you can separate the root rote pathogens from the others???

Answer: At line 199 are described the effects of Gnomoniopsis castaneae attack and its first discover. Since that this fungus is still at the first stages of studies, little is still known about its epidemiology (included the ways in which inoculum can be transported and the species that this fungus can colonize as endophyte). However, across the literature we haven’t found scientific evidence of the capability of this disease to spread from Chestnut to other species. With respect to Ceratocystis fimbriata, the species Ceratocystis platani in Italy it is widespread among Plane trees but it is a species that has been reported only on Plane species. In addition, no reports of Ceratocystis spp. affecting Chestnut species were found among the analyzed literature.

Rev 2: Maybe  creating a Table specific to chestnut  for each  headings with name of pathogens or process etc... and  linked literature will be useful for the readers. Although, I understand that due to copyrights  regulations, it is difficult to insert some photos, It could be valuable to include some photos which are free of charges in order to illustrate some of the aspects of this review.

Answer: we warmly thank reviewer 2 for his valuable suggestion. With respect to the insertion of photos or illustrations in the paper, we definitely agree with reviewer 2; in particular, the journal sustainability is an open access journal, so we do not have the permission to re-publish (in open access) photos or other materials with copyright.

With respect to tables, we also agree with reviewer 2, but, unfortunately, we tried to create tables which summarized the content but their insertion in this review led to the exceeding of word limit count; so we couldn’t add in the paper.

However, I’m glad that all the tree reviewers judged the paper as “easy to read” and “an easy reading paper”.

Rev 2: Overall, congratulations to the authors as this is an extensive well focused review.

Answer: Thanks. We are glad that our review is appreciated by reviewer 2.

Reviewer 3 Report

The manuscript From chestnut tree (Castanea sativa) to flour and foods: A systematic review of the main criticalities and control strategies towards the relaunch of chestnut production chainsummarize current knowledge regarding the main criticalities in chestnut tree cultivation, chestnuts processing, and in chestnut flour production,and suggest specific improvement strategies to contrast the main pests and diseases affecting chestnut tree, improve chestnut processing and flour production. This manuscript summarized that the recovery and valorization of nutraceutical components (mainly polyphenols) from chestnut by-products is the most interesting, sustainable, and profitable strategy. However, the language of the paper is not pretty good and needs improvements, the authors should carefully check the version with appropriate expression to meet the academic requirements. Thus, the manuscript can be accepted to publish in the journal with modifications.

1.     L44, “[4].     C. sativa is considered as….”  The space between [4] and C. sativa should be deleted.

2.     L69-70, L71-74, the sentences is poor English, please rephrase.

3.     L184-189, there are some grammar mistakes in the sentences, please correct them.

4.     L288, “The possibility of determine the presence.......” , determinein the sentence is not very suitable。

5.     L338-340, the sentence is too long and also have grammar mistake, please check it.

6.     L465, “Chestnuts can be contaminated by molds before their harvest and not only”, the meaning of the sentence is very confused, please check it.

7.     L712-714, the sentences is poor English, please rephrase.

8.     L731, the space between “that” and “A. japonicas” should be deleted.

Author Response

Dear Editor,

the authors would like to thank reviewer 3 for his valuable comments. Here following, you can find a reply point by point to the reviewer suggestions.

Rev 3: The manuscript “From chestnut tree (Castanea sativa) to flour and foods: A systematic review of the main criticalities and control strategies towards the relaunch of chestnut production chain” summarize current knowledge regarding the main criticalities in chestnut tree cultivation, chestnuts processing, and in chestnut flour production, and suggest specific improvement strategies to contrast the main pests and diseases affecting chestnut tree, improve chestnut processing and flour production.

This manuscript summarized that the recovery and valorization of nutraceutical components (mainly polyphenols) from chestnut by-products is the most interesting, sustainable, and profitable strategy. However, the language of the paper is not pretty good and needs improvements, the authors should carefully check the version with appropriate expression to meet the academic requirements. Thus, the manuscript can be accepted to publish in the journal with modifications.

  1. L44, “[4]. C. sativa is considered as….”  The space between [4] and C. sativa should be deleted.

Answer: done, thanks. Please see line 42

Rev 3.  2.   L69-70, L71-74, the sentences is poor English, please rephrase.

Answer: done, thanks. Please see line 69–76

REV 3: 3.     L184-189, there are some grammar mistakes in the sentences, please correct them.

Answer: done, thanks. Please see lines 182–187.

Rev 3: 4.     L288, “The possibility of determine the presence.......” , “determine” in the sentence is not very suitable。

Answer: we agree with reviewer 3. The sentence has been rephrased. Please see line 286

Rev 3: 5.     L338-340, the sentence is too long and also have grammar mistake, please check it.

Answer: done, thanks. Please see lines 336 – 339

Rev 3: 6.     L465, “Chestnuts can be contaminated by molds before their harvest and not only”, the meaning of the sentence is very confused, please check it.

Answer: thanks for the comment. The second sentence (463-464), explain the statement “not only” reported in the first sentence.

Rev 3: 7.     L712-714, the sentences is poor English, please rephrase.

Answer: done, thanks. Please see lines 710 –712

Rev 3: 8.     L731, the space between “that” and “A. japonicas” should be deleted.

Answer: done, thanks. Please see line 729.
